# Widespread natural methane and oil leakage from sub-marine Arctic reservoirs

Pavel Serov [1] ✉, Rune Mattingsdal[2], Monica Winsborrow[1], Henry Patton [1] & Karin Andreassen[1]

Parceling the anthropogenic and natural (geological) sources of fossil methane in the atmosphere remains problematic due to a lack of distinctive chemical markers for their discrimination. In this light, understanding the distribution and contribution of potential geological methane sources is important. Here we present empirical observations of hitherto undocumented, widespread and extensive methane and oil release from geological reservoirs to the Arctic Ocean. Methane fluxes from >7000 seeps significantly deplete in seawater, but nevertheless reach the sea surface and may transfer to the air. Oil slick emission spots and gas ebullition are persistent across multi-year observations and correlate to formerly glaciated geological structures, which have experienced km-scale glacial erosion that has left hydrocarbon reservoirs partially uncapped since the last deglaciation ~15,000 years ago. Such persistent, geologically controlled, natural hydrocarbon release may be characteristic of formerly glaciated hydrocarbon-bearing basins which are common across polar continental shelves, and could represent an underestimated source of natural fossil methane within the global carbon cycle.

Release of fluids (liquids and gases) such as methane, from the seafloor is a consequence of diverse chemical, biological, geological and physical processes occurring in the underlying strata, and is widespread and multifarious across continental margins[1]. The magnitude of seafloor methane ($CH_4$) release reflects the balance between methane generation and its microbial degradation, and between its buoyancy-driven disposition to migrate upwards and the ability of geological layers to retain it. Due to its ability to drive global warming 28–36 times more effectively than carbon dioxide over a period of 100 years[2], natural and anthropogenic methane dynamics in the Earth interiors, oceans and the atmosphere have been a research focus for several decades[3]. Current rates of increase of atmospheric methane concentration ($9.3 \pm 2.4$ ppb yr$^{-1}$ during the period 2014–2019) are predicted to enhance atmospheric warming over decadal timescales, hindering efforts to stay below the global temperature targets of the Paris Agreement[4,5]. Yet, a major gap between global methane budgets estimated from inverse modeling (top-down) and from empirical upscaling of point source measurements (bottom-up) demonstrate

the incomplete understanding of global methane sources and sinks[3,5–8]. Past and present atmospheric methane emissions remain subject to debate largely due to difficulties partitioning contributions of different sources in atmospheric gas records[9]. Discriminating the sources of fossil $^{14}C$-free methane (~30% of global methane budget[10]) such as emissions during hydrocarbon extraction, coal mining, marine and terrestrial natural seepage from hydrocarbon basins, etc. is particularly challenging due to often similar $\delta^{13}C$ and $\delta D$ composition of the emitted $CH_4$[3,7,11]. Because atmospheric data alone do not provide source type information for these isotopically alike emissions, mapping and quantifying the $CH_4$ inventories and their dynamics is necessary to improve partitioning between anthropogenic and natural sources of fossil methane[8,12].

Both top-down and bottom-up approaches to methane budgeting traditionally attribute marine methane sources (including but not limited to submarine clathrates) to microbially mediated degradation of organic matter in shallow sediments (microbial gas)[13–18]. In situ point source measurements across the ocean floor show that the number of

[1]CAGE-Centre for Arctic Gas Hydrate, Environment and Climate, UiT–The Arctic University of Norway, Tromsø, Norway. [2]NPD—Norwegian Petroleum Directorate, Harstad Office, Harstad, Norway. ✉e-mail: pavel.russerov@uit.no

observed microbial methane seeps is greater than thermogenic, which originate from thermal cracking of buried organic matter[19,20]. Yet data have not been collected systematically and remain scarce, particularly in the Arctic. Owing to this paucity of empirical data, the most recent gridded 1° x 1° methane emission maps reveal only three submarine seep areas in the Arctic, all of which emit predominantly microbial gas[19]. Two of these are shallow permafrost-bearing shelves off East Siberia and Alaska[21,22], whilst the third corresponds to the shallow shelf (80–240 m water depth) on the western Svalbard margin[23,24] (Fig. 1). Seepage of thermogenic $CH_4$ may lack representation in global emission estimates, in particular when it is considered that 33 million km² of the Arctic shelf area contain hydrocarbon reserves[25–27] and has experienced successive highly erosive Quaternary glacial cycles[28–30] that are known to promote fluid discharge at the seabed[31,32] through unsealing of petroleum reservoirs, fault permeability changes[33,34], tilting of reservoirs[35,36], and pressure-driven expansion of gas and eventual seal failures[37]. Furthermore, leaking petroleum-bearing basins are prone to the concurrent release of both oil and gas, potentially making such settings more efficient for the transportation of $CH_4$ to the sea surface than just gas seepage sites, as oil-coated gas bubbles are thought to be less susceptible to dissolution and, thus, to microbial degradation in the water column[38,39].

The Barents Sea shelf represents one such Arctic shelf, with a complex history of uplift and erosion resulting in the removal of 1.7–2.6 km of overburden over a period from ~50 Ma BP until the end of the last glacial cycle ~20 ka BP[40,41], combined with substantial 2966 Sm³ mill. o.e. (million standard cubic meters oil equivalent)[42] discovered and undiscovered hydrocarbon resources within the Norwegian sector alone.

The Barents Sea consists of an intricate array of basins hosting a near-continuous sedimentary succession from the Carboniferous to Quaternary, and structural highs bearing fragmented sedimentary succession due to episodes of erosion[27,30]. The northern Norwegian Barents Sea bears a suite of >10 km wide and 50–100 km long anticline structures within the Kong Karls platform and two structural highs—Sentralbanken high and Storbanken high (Fig. 1) all of which were grabens or rift-bounded basins in the Late Paleozoic. Subsequent compression movements initiated in Late Jurassic inverted these basins/grabens[27] exhuming older Mesozoic and Paleozoic strata, including Triassic and Jurassic hydrocarbon source and reservoir units[27,42].

The hydrocarbon potential of the northern Norwegian Barents Sea is estimated to be significant[42]. However, traces of hydrocarbons are frequently found in "dry" wells and surface sediments suggesting hydrocarbon loss due to widespread paleo fluid leakages[31,43]. Resonating with geochemical observations, basin modeling results[33] suggest that the sealing potential of hydrocarbon reservoirs may be compromised due to regional uplift and 850–1370 m net erosion in Paleogene and Neogene[40,44] and 940 to 1180 m net glacial erosion during >40 reciprocal glaciations in Pleistocene[41].

Across the northern Norwegian Barents Sea, the main source rock with confirmed hydrocarbon potential is the Lower–Middle Triassic Steinkobbe Formation, which consists of organic-rich (2.4–10% total organic carbon) marine and delta front fine-grained deposits[45,46]. This source rock is overlain by at least four potential reservoir units: Kobbe, Snadd, Tubåen, and Stø Formations–all deposited during early Mesozoic infilling of the Barents Sea basin with coastal, deltaic and shallow marine terrigenous deposits[45] (Fig. 2). In contrast to the reservoir formations, deposition of low permeability seal units across the Barents Sea was limited. The most prominent seal is the Upper Jurassic Hekkingen and Fuglen Formations which cover the entire succession of Triassic and Jurassic reservoirs (Fig. 2). Upper Triassic marine shales of the Norian Flatsalen Formation, Lower Ladinian shales, and sporadic Upper - Middle

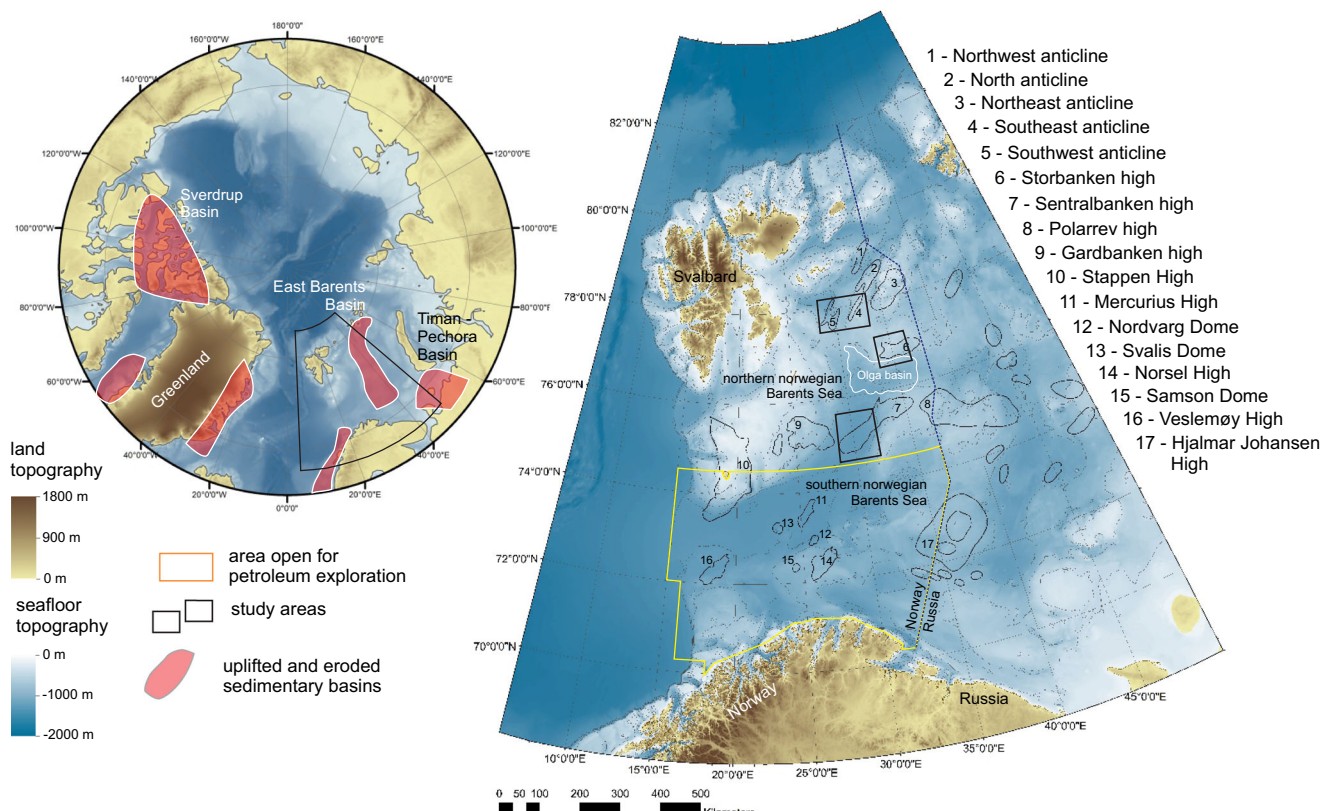

1 - Northwest anticline
2 - North anticline
3 - Northeast anticline
4 - Southeast anticline
5 - Southwest anticline
6 - Storbanken high
7 - Sentralbanken high
8 - Polarrev high
9 - Gardbanken high
10 - Stappen High
11 - Mercurius High
12 - Nordvarg Dome
13 - Svalis Dome
14 - Norsel High
15 - Samson Dome
16 - Veslemøy High
17 - Hjalmar Johansen High

**Fig. 1 | Structural elements[42] of the Barents Sea in the context of tectonically uplifted and glacially eroded Arctic sedimentary basins[30,64,66,68,70,83], with the naming conventions following Norwegian Stratigraphical Committee.** Bathymetric maps are based on IBCAO V4 data[84].

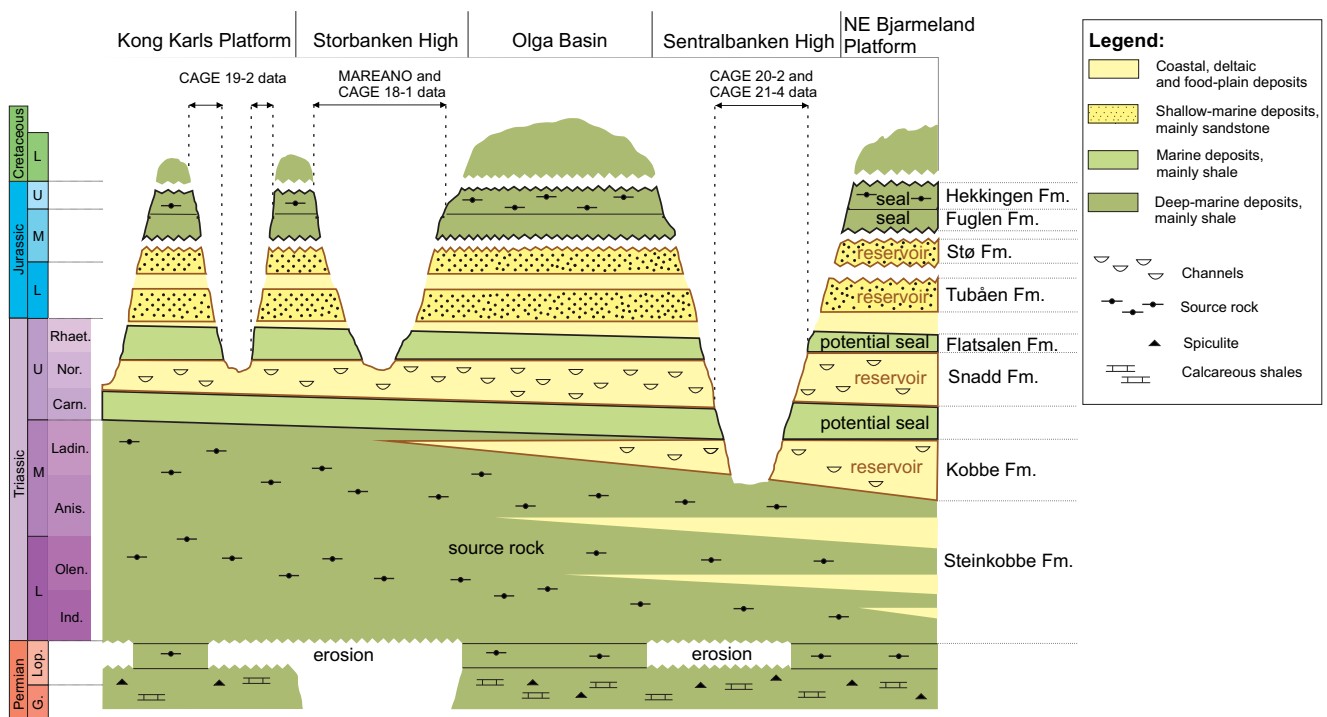

**Fig. 2 | Lithostratigraphic diagram for the northern Norwegian Barents Sea shelf**[42]. Bold black lines outline potential reservoir formations, bold brown lines outline reservoir units. The erosional topography and thickness of the lithostratigraphic units are not to scale.

Triassic transgressive marine shale beds may also constrain fluid migration.

Here we present seismic data, water column echosounder imaging of gas release, methane concentration measurements in seawater, and remote sensing datasets that demonstrate extensive hydrocarbon leakage from the subseafloor through the water column and to the sea surface, associated with petroleum systems within the northern Norwegian Barents Sea, Arctic Ocean. These data document one of the largest cold seep regions in the Arctic, and also evince the natural oil seepage in the Barents Sea. Our empirical data point to a geological setting that is conductive for strong thermogenic hydrocarbon leakage. Hypothesizing such settings may be pervasive across previously glaciated and extensively eroded Arctic continental shelves, we engage two additional less diverse surveys to test whether other geological structures reminiscent to our main study site in Sentralbanken high also emit hydrocarbons (Fig. 1). Observed methane dynamics in the water column across three study sites suggests that naturally uncapped hydrocarbon reservoirs may contribute to methane budgets within ocean water and, possibly, atmosphere and need to be accounted for in Arctic marine methane source estimates.

## Results and Discussion
### Oil and gas in seawater
At Sentralbanken high we identified 4,137 acoustic 'flares' diagnostic of bubble emission sites (seeps) along discrete multibeam swaths. The total seafloor coverage of the water column multibeam data for gas flare mapping is ~660 km². Overlapping echosounder swaths circumstantially acquired at different tidal settings reveal no corelation between tidal cycle and gas flare abundance and their relative strength (Fig. S1).

Data reveal a strongly heterogenous spatial distribution of the flares, where the majority of the population is grouped in 1 km² to >7 km² distinctive clusters with sharp boundaries (Fig. 3). Solitary flares, which do not have any neighbours within a 500 m radius, comprise only 2.7% of the total flare number. In plan view, the seepage clusters appear as isometric patches unevenly distributed across the

structural high with the most pronounced flare clusters graviating towards its central part. We did not observe any elongated strings of flare reminiscent of fault lineamets[47,48] or clusters following certain bathymetric contours as would be expected if the edge of a gas hydrate stability zone defined the gas release[49,50]. The central part of the Sentralbanken structural high demonstrates the highest gas flare density (Fig. 4a). Here, the flare clusters with clear-cut boundaries show maximum flare densities exceeding 100 seeps per km². These clusters also contain higher ratios of strong and moderate flares (>50% in total) compared to other zones of the Sentralbanken high, where weak flares comprise the considerable majority of flare population (Fig. 4). Across the entire data set, we classified 620 strong flares, 1384 medium flares, and 2133 weak flares, defined by their apparent backscatter strength and height reached within the water column (Fig. 3e). Based on the observed spatial distribution of flare clusters and their semi-quantitative characteristics (flare density and proportions of strong, moderate and weak flares), we outline three seepage zones (Fig. 4a). The seepage zones demonstrate a NE orientation, similar to the dominant orientation of the regional structural elements: Sentralbanken high, Storbanken high and elements of Kong Karls platform (Fig. 1).

Outside the most pronounced seepage zones, flare clusters gradually become rarefied and their margins less well defined (Fig. 4c). Within such rarefied, patchy flare clusters, the seep density does not exceed 60 flares * km⁻² and weak flares compose >76% of the total population, consistent with more obstructed advection of fluids through strata or a weaker gas source (Figs. 3, 4).

89 flares, in addition to producing a very strong backscatter signal, also appear to reach close (<50 m) to sea surface before terminating or leaving the echosounder footprint (Fig. 3). Because the height of the insonified water column decreases with increasing swath angle (see Methods), detecting accurate termination of such high flares is only possible within a narrow corridor directly beneath an echosounder, even with 50% swath overlap. This suggests that the actual number of seeps reaching the upper water column section may be substantially larger.

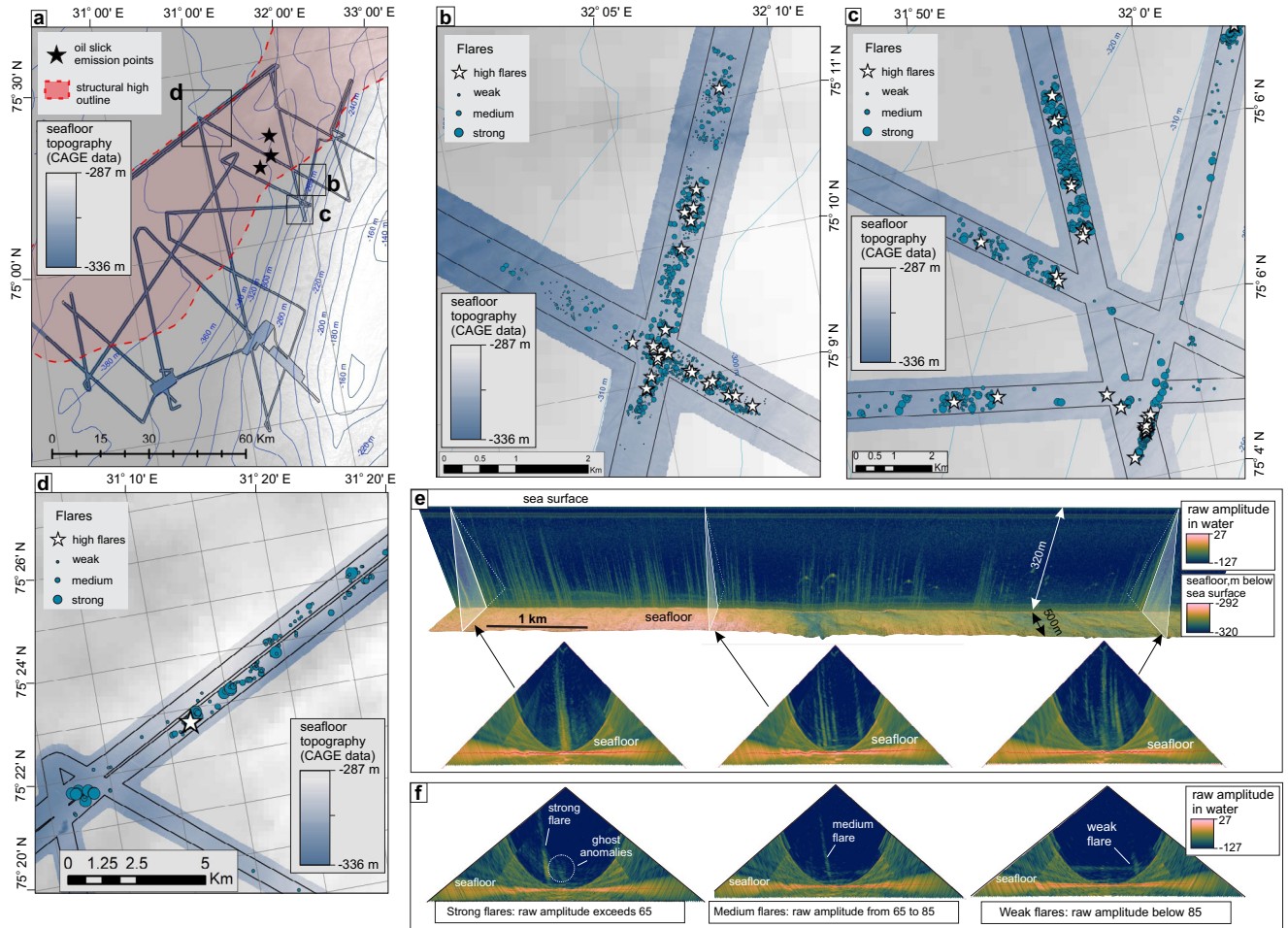

**Fig. 3 | Imaging and mapping of the gas release at Sentralbanken high.**
**a–d** Locations of gas flares at Sentralbanken high. Multibeam bathymetric data is shown in chromatic colour scale. IBCAO V4 bathymetric data[84] is shown in mono-chromatic colour scale. Black lines delineate coverage of data suitable for gas flare mapping. **e** Gas flares on multibeam echosounder data in stack view (top) and fan view (bottom). **f** Examples of strong, medium, and weak acoustic gas flares. Source data are provided as a Source Data file.

Examining synthetic aperture radar (SAR) images over a 7-month period, we identified persistent oil slicks on the sea surface in the Sentralbanken region (Fig. 5a, c). These could be linked to three oil emission sources located at the edge of seepage zone 1 where gas release is somewhat tempered compared to the most intense seepage zones 2 and 3 (Fig. 4). The shape and orientation of the surface slicks varied due to surface current conditions (Fig. 5c). Typically, the slicks formed 7–10 km long and 0.2–1 km wide stripes and were easily visible on sea surface during the CAGE 21-4 research cruise (Fig. 5b).

In line with abundant acoustic evidence of seafloor gas release, water samples indicate that the entire water column is supersaturated with methane. All collected seawater samples show high (35 to 752 nM) concentrations of dissolved $CH_4$ in the bottom water layer and in the intermediate water interval from 50 m above the seafloor to 50 m below the sea surface (8.5 to 22.8 nM). Notably, all water samples from the surface mixed layer reveal $CH_4$ concentrations that exceed the sea water−air equilibrium suggesting that $CH_4$ is diffusing to the air (Table 1). Given that the seabed seepage is continuously replenishing the seawater $CH_4$ pool across a ~40 × 70 km area, the cumulative input of this potential $CH_4$ source deserves attention. It is important to emphasize that despite apparent contribution of methane from sea-floor seepage, our water column methane analyses cannot rule out some contribution of methanogenesis in the oxic water column[51,52].

The composition of hydrocarbon gas in sediments reveals a fraction of C2-C5 hydrocarbon gas (ethane, propane, etc.) indicative for a thermogenic origin of gas[1]. All samples ($n = 36$) collected from four sediment (Fig. 5a) contained 0.3% to 6.8% ethane, 0% to 2.5% propane, as well as a suite of less abundant heavier homologs. Core GC 347 collected at the deepest (360 m water depth) zone of the study area from a 5 m tall and 120 m diameter seafloor mound structure revealed gas hydrates. Other cores collected at 301–330 m water depth did not contain hydrate, nor did they reveal sediment textures (e.g., moussy sediment) circumstantially evident for recent gas hydrate decomposition.

## Geological control of fluid release

Sub-seafloor geology appears to rigidly control the clustering of gas flares. The apex of the Sentralbanken high bears several anticline structures with eroded tops, exposing reservoirs of the Middle Triassic Kobbe Formation and providing 2–5 km wide 'windows' for uncon-strained fluid discharge (Fig. 6). Each of these 'windows' corresponds to a prominent seepage zone with high density of seeps (Figs. 3, 4a, 6). Despite the presence of bright spots indicative of fluid accumulations in the sub-seafloor scattered across the entire apex of the Sen-tralbanken high, strong seabed fluid release is restricted to those parts with subcropping reservoir formation (Fig. 6). The uppermost Kobbe Formation reservoir is connected to deeper reservoirs of Klappmys and Havert Formations by faults and may therefore act as a transitional capacitor for fluids originating from deeper strata, including Upper Paleozoic levels.

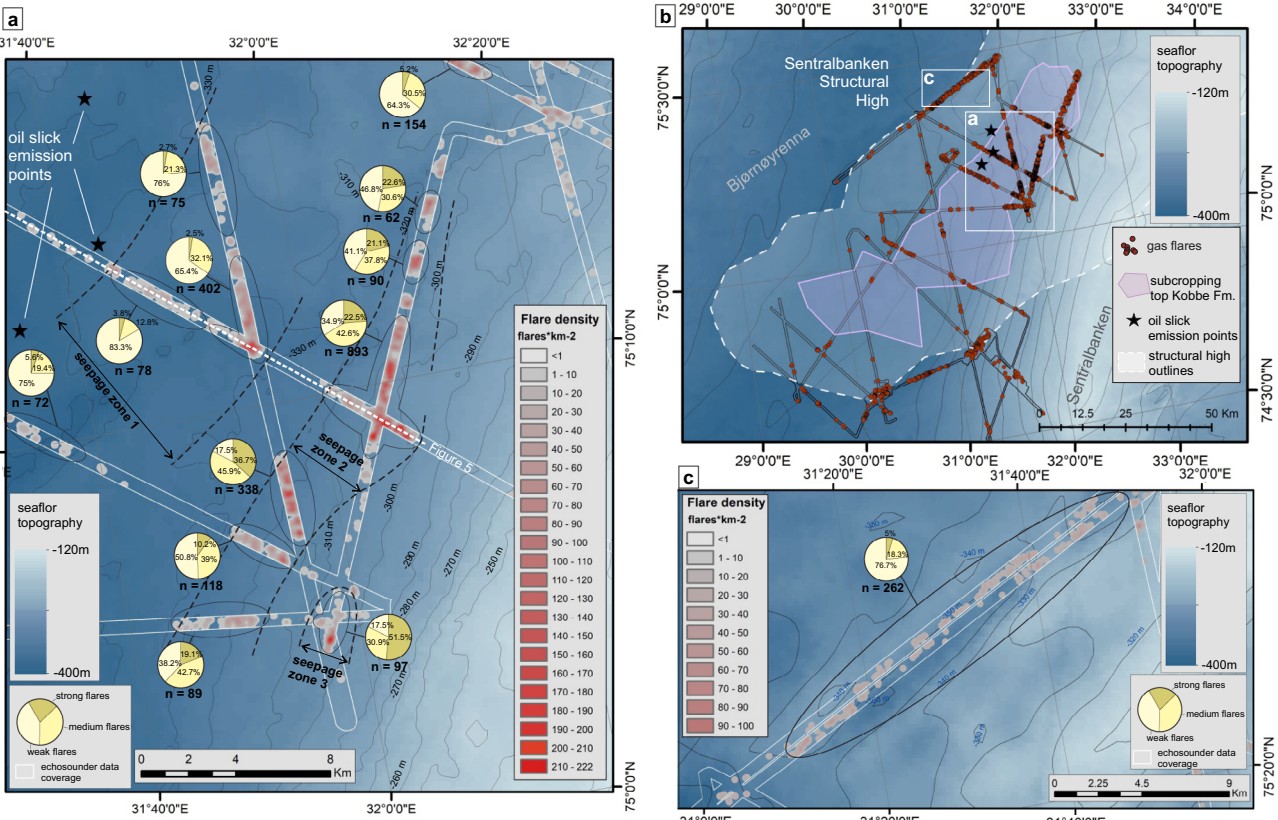

**Fig. 4 | Density of gas flare appearance and ratios between strong, moderate and weak flares.** Multibeam bathymetric data are shown in monochromatic color scale. IBCAO V4 gridded bathymetric data[84] are shown in blue color scale. **a** Central part of the Sentralbanken high. **b** Flare distribution within entire data set from Sentralbanken high. **c** Flare densities along the compressional fault zone. Source data are provided as a Source Data file.

On the flanks of the structural high where the overburden remains and the reservoir formation is tapered compared to its apex (Fig. 6), several small flare clusters with a seep density <50 seeps* km$^{-2}$ correlate to the faults piercing down to the top Kobbe Formation. In such settings all seepage correlates to faults.

Within the apex area of the Sentralbanken high where the reservoir formation reaches its maximum thickness and accommodates scattered bright spots, normal faults potentially promoting fluid migration toward the seafloor are abundant. However, there is no consistent correlation between seep intensity and faulting, with the highest seep densities (>100 seeps*km$^{-2}$) not always associated with faults visible on our seismic data (Fig. 6). Furthermore, we observe no apparent correlation between bright spot distribution below the seabed and the location or magnitude of seabed release of gas, with bright spots occurring beneath anomalously strong leakage areas, as well as areas with temperate and no leakage. It is therefore not clear whether faulting plays a significant role in modulating seepage at exhumed and partially subcropping reservoirs that are already preconditioned for strong degassing through erosion. However, on the flanks of the structure where overburden remains, faulting appears to be the dominant mechanism driving tempered seabed gas release.

A veneer of Quaternary unlithified sediments is not detectable on the conventional seismic profiles (-20 m vertical resolution at the seafloor) available from the study area. However, higher resolution sub-bottom profiler data document a ~3–10 m thick layer of glacigenic deposits (Fig. S2). Furthermore, occasional iceberg plough marks, meltwater channels and pockmarks visible on multibeam bathymetry data are consistent with a drape of unlithified deposits covering the lithified Mesozoic sedimentary strata. However, we do not find correlation between the presence of pockmarks and seabed seepage (Fig. S3).

## Methane leakage from other eroded structural highs

To test whether the strong hydrocarbon discharge identified from the eroded structural high setting of Sentralbanken is a circumstantial phenomenon, or whether hydrocarbon leakage may be expected in similar geological settings elsewhere across the formerly glaciated Barents Sea shelf, we investigated the relation between water column gas flares and the structural boundaries of the Storbanken high and anticline structures within the Kong Karls platform, north of Sentralbanken high (Fig. 7). Multibeam echosounder datasets from Kong Karls platform (a discrete line survey), and Storbanken High (2810 km$^2$ areal data set acquired by the MAREANO seabed mapping project) both reveal intensive gas flaring confined to emerged structural elements (Fig. 7). At Kong Karls platform, SW and SE anticlines expose the Snadd Formation at the seafloor in their core parts[42] (Fig. 2), and correlate closely to 597 gas flares. In Storbanken high, 2646 flares within an 2810 km$^2$ area are identified, and the most distinct flare strings correlate to subcropping upper Triassic Snadd Formation and lower-middle Jurassic Tubåen and Stø Formations[42,53] (Figs. 2, 7).

## Capping effect of overburden

Our datasets from across the Central Barents Sea reveal 7380 hydrocarbon gas seeps originating from eroded structural highs bearing thermogenic hydrocarbon sources. Aiming to address whether such extensive hydrocarbon gas venting may be significant in the context of global marine methane seepage, we compare our gas flare mapping results with other pronounced marine seep regions that have been surveyed with echosounder systems.

Extensive gas flaring has been documented along the continental margin off Svalbard and along the western margin of the Barents Sea shelf from Bear Island to Kongsfjorden[23,47,54]. Data was acquired with Kongsberg EM710 multibeam system. Although, the exact insonified

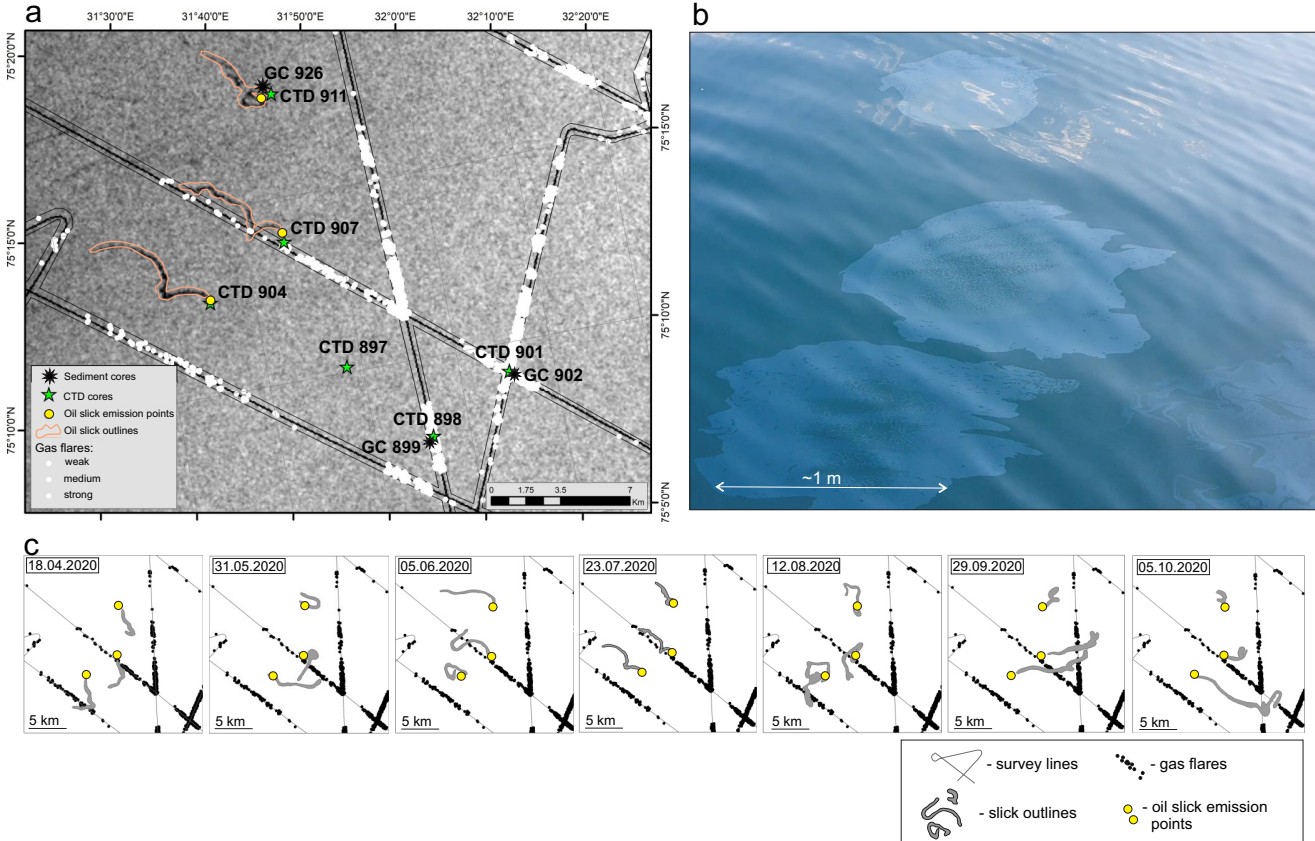

**Fig. 5 | Indications of methane gas and oil in sea water. a** Synthetic aperture radar (SAR) image s1b-ew-grd-hh-20200723t051748 (Copernicus Sentinel data 2020, processed by ESA) of oil slicks on sea surface and locations of gas flares, water sampling stations, and sediment cores. **b** Photograph of oil films taken during oil sampling in the vicinity of CTD 911 during CAGE 21-4 research cruise. **c** Compilation of monthly oil slick observations from April to October 2020. Source data are provided as a Source Data file.

**Table 1 | methane concentration in the surface mixed layer and the methane fluxes from the sea to the air**

| Sample | CH$_4$ concentration in surface mixed layer (5 m below the sea surface), nmol l$^{-1}$ | Percent saturation | Wind speed 10 m above the sea surface, m s$^{-1}$ | Sea-air flux, µmol m$^{-2}$ d$^{-1}$ |
|---|---|---|---|---|
| CTD 897 | 4.2 | 135 | 6.5 | 2.7 |
| CTD 898 | 3.7 | 120 | 6.4 | 1.5 |
| TD 901 | 4.1 | 132 | 7.2 | 3.0 |
| CTD 904 | 3.8 | 122 | 5.5 | 1.2 |
| CTD 907 | 4.2 | 135 | 4.8 | 1.4 |
| CTD 911 | 3.8 | 124 | 7.4 | 2.4 |

area is unknown, >1000 gas flares occur within a ~600 × 50 km stretch of this continental margin and fuel a large plume of dissolved methane in the water column[47]. The pronounced Hornsund Fault Zone has been hypothesised to control subseafloor fluid migration, while shrinkage or expansion of the gas hydrate capacitor driven by seasonal water temperature fluctuations[23,49] may further mediate seabed release. However, the shallowest seep clusters are located at 90 and 240 m water depth and are therefore significantly outside of the gas hydrate stability envelope. The stable isotope composition of emitted methane and the absence of heavier methane homologs indicate a microbial gas origin[47,55,56].

Along the northern US Atlantic margin, ~570 seeps were identified within 94,000 km² area covered with multibeam echosounder data at 50–1700 m water depth, ~440 of which originate at water depths bracketing the upper limit of the gas hydrate stability zone[50], similar to parts of the western Svalbard margin[23,49]. This study and our study use the same EM302 echosounder model. The origin and migration pathways of the gas are not clear, yet dynamics of the upper gas hydrate stability zone limit is hypothesized to control seep occurrences[50].

In contrast to the northern US Atlantic margin and the continental margin west of Svalbard, at Sentralbanken high, the gas seep clusters crosscut bathymetric contours and appear across a wide range of water depths, which is not characteristic for gas release fuelled by the retreat of a gas hydrate layer. However, recovery of gas hydrates at 360 m water depth show that the deepest SW parts of the structural high may be within the zone of gas hydrate stability. We note that seabed gas release is less pronounced at this deeper region, yet it also lies outside of the eroded structural top. At the apex of Sentralbanken high, the absence of a seismic bottom simulating reflector and no confirmed gas hydrate recoveries, together with very extensive and geologically constrained seepage (Fig. 6) suggest that there is no gas hydrate layer capping natural degassing of petroleum reservoir today. At Storbanken high and Kong Karls platform, shallower (~100–290 m) water depths point to very limited potential to host pressure and temperature-dependent gas hydrates.

In continental margin settings off Western Svalbard and the northern US Atlantic margin, the sedimentary overburden may limit fluid release[24,50]. However, across the uplifted, eroded, and repeatedly glaciated Barents Sea shelf, the sedimentary rock overburden has been significantly reduced[41], and only a thin veneer of marine and glacigenic deposits of the last glacial-interglacial cycle overlay the erosional surface of the Mesozoic sedimentary succession. The reduced capping effect of the overburden and limited capping effect of gas hydrates,

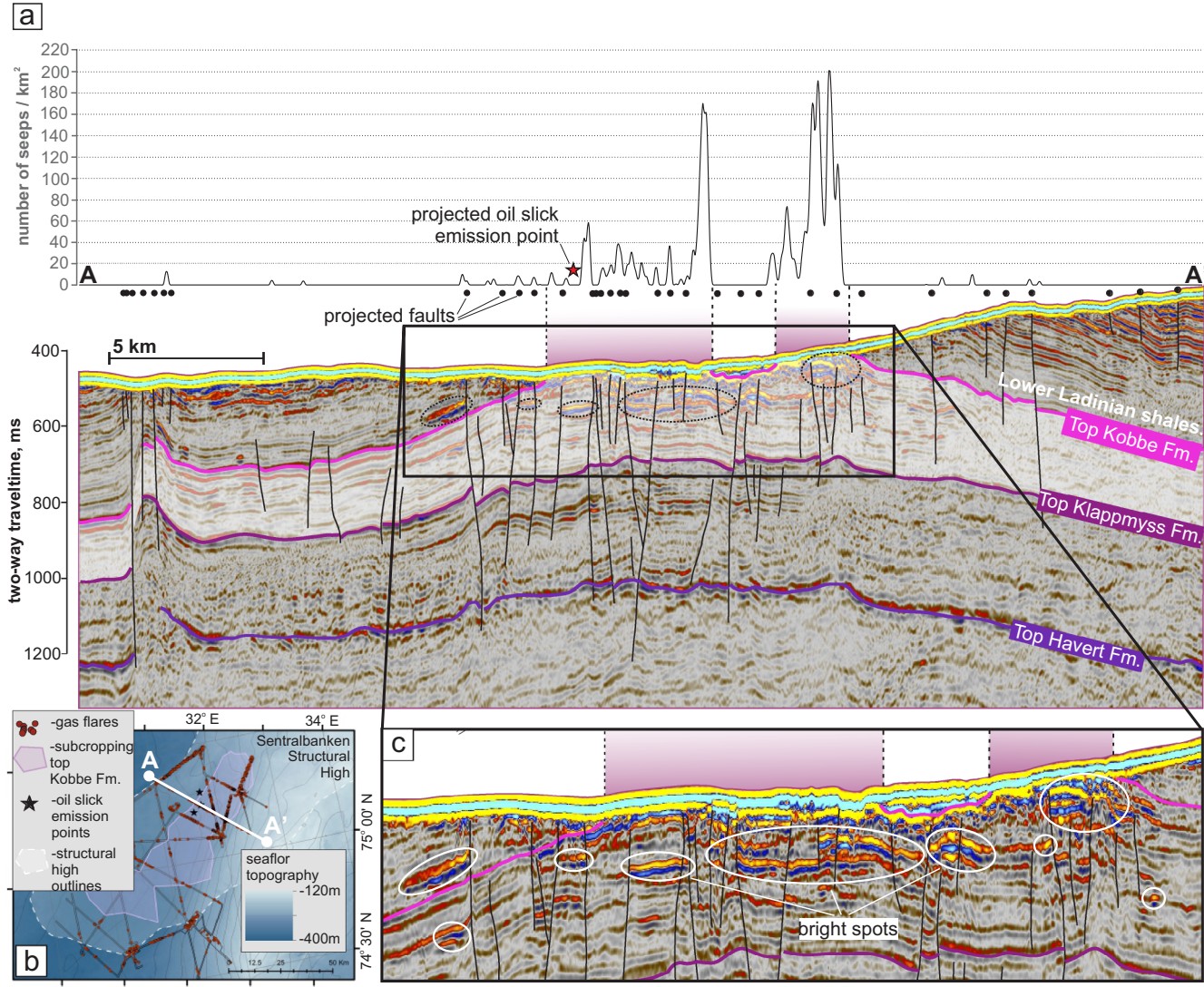

**Fig. 6 | Correlation of gas release and subcropping reservoir Kobbe Formation and faults. a** Seepage density, faults and subcropping Kobbe Formation along transect A - A'. White shading shows the reservoir. Black subvertical lines show major faults. **b** location of transect A - A'. **c** Distribution of bright spots within the apex of Sentralbanken structural high. Source data are provided as a Source Data file.

together with hydrocarbon abundance in the shallow subsurface, provide the ideal geological setting for extensive seepage. Based on the timing of the last Barents Sea ice sheet collapse[57]–the last major erosional event to affect this region–hydrocarbon leakage associated with the degradation of the overlying overburden seal has been possible, and potentially ongoing, for the last c. 15,000 years, at least.

At the three Barents Sea sites 7380 seeps have been identified within 3730 km² of surveyed area (Fig. 7). This density of seepage must significantly exceed the ~1000 seeps within 30,000 km² of the Western Svalbard margin[47] and the 570 seeps within 94,000 km² of the northern US Atlantic margin[50]. Given limited water column data coverage (Fig. 7, Fig. S3), we can confidently assume that the actual number of seeps within investigated structures in the northern Norwegian Barents Sea must be substantially higher, making this one of the most active submarine methane release hotspots globally.

## Methane dynamics in seawater

Because free methane gas is not subject to microbial degradation in the water, bubble transport through the water column is a potent delivery mechanism to the atmosphere[39]. Methane from marine seep sources has been shown liberating to the atmosphere at shallow shelf

settings (20–50 m water depth at Santa Barbara Channel offshore California[58], <50 m water depth on the East Siberian Arctic shelf[59], 50–120 m water depth offshore Svalbard[60], etc.) where gas bubbles reach close to the sea surface before losing their $CH_4$ content due to mass transfer with seawater. In deeper water settings, the longer exposure of bubbles to sea water decreases the transport efficacy.

In Sentralbanken area we encountered a water column supersaturated with respect to atmospheric methane equilibrium at all sampling stations and at all water levels, pointing towards the expansion of the methane plume across the actively seeping area and throughout the water column. Surface mixed layer $CH_4$ concentrations produce sea to air flux ranging from 1.2 to 3 µmol m⁻² d⁻¹, which is lower compared to 2 to 20 µmol m⁻² d⁻¹ flux reported at shallow seep regions offshore Svalbard[56,60]. The fluxes are transient because the oceanographic conditions (e.g., water column stratification and current strength) control methane plume dynamics and the wind speed affects sea-air gas transfer significantly. It is interesting to note that the Sentralbanken study site has been circumstantially investigated in terms of methane mixing ratios in air prior to our discovery of seabed gas and oil seepage[61]. Methane mixing ratios above the structural high were found to exceed 2000 ppb during the autumn and winter months[61]

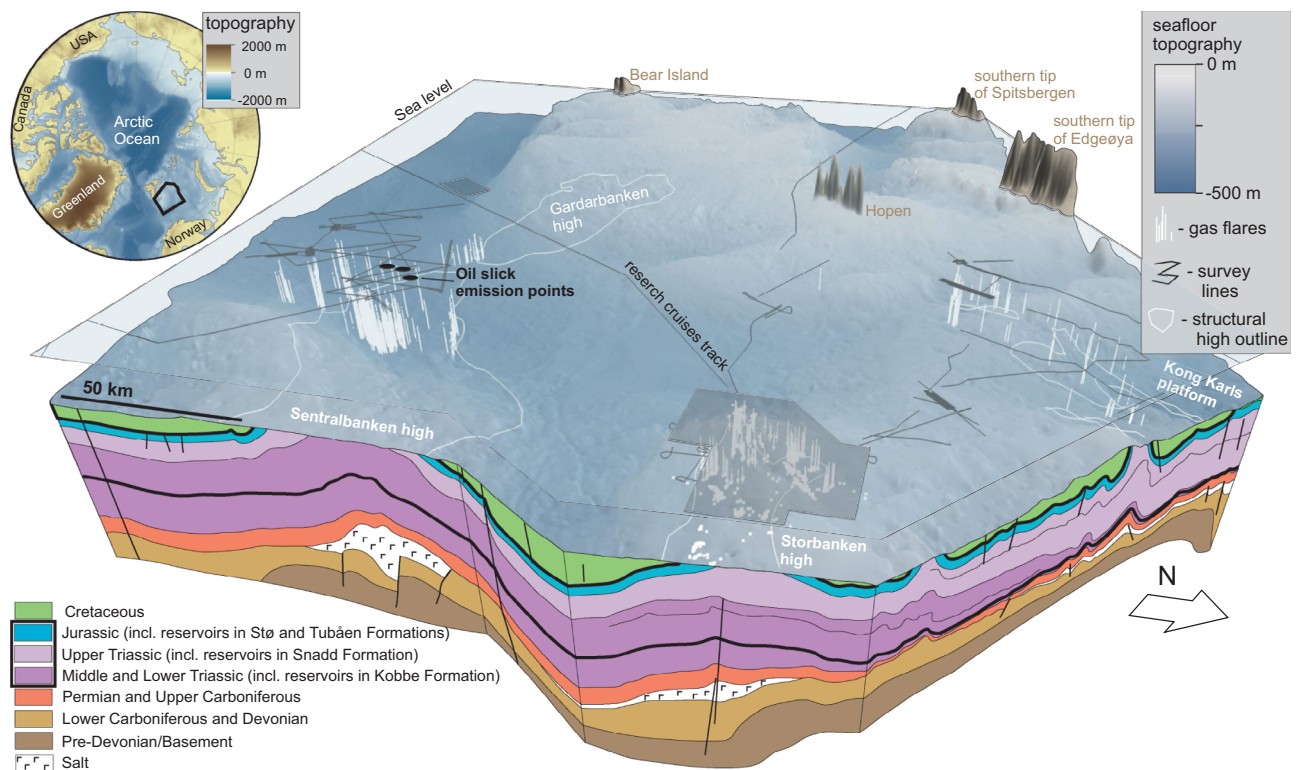

**Fig. 7 | Exhumed hydrocarbon-bearing structures in the northern Norwegian Barents Sea emitting gas and oil to the water column.** Subvertical white stripes and white dots show gas seeps. Shaded polygons on the sea surface show aerial multibeam echosounder data coverage. White polygons indicate structural boundaries. The geological profile (modified) is based on NPD, Geological assessment of petroleum resources in eastern parts of the Barents Sea North. (2017)[42]. Bold black lines on the geological profile highlight Triassic and Jurassic hydrocarbon reservoir-bearing strata. Bathymetric data is from IBCAO V4[84].

(the background methane mixing ratio is ~1950 ppb). Platt et al.[61] interpreted the excursions from baseline methane mixing ratio as long-distance transport from land-based sources, however, our findings should motivate re-examination of atmospheric methane concentrations in this area.

The simultaneous release of both methane gas and oil at Sentralbanken high, may contribute to the extreme size and density of free methane gas transport in the water column at this site. Oil coating of methane bubbles is known to decrease mass transfer between the bubbles and the water column[62]. The presence of persistent oil slicks within the seepage area and observations of oil droplets reaching the sea surface during sampling of these slicks, indicate that oil is leaking from the seafloor and reaching the sea surface. However, our methane concentration data collected 5 m below the oil slicks, did not show higher methane content compared to surface water samples collected outside the slicks (Fig. 6). Video surveys of the seafloor with remotely-operated vehicle confirm that some of the seeps are composed of oil-coated bubbles. However, gas ebullition on sea surface has not been observed and none of gas flares reach the sea surface, despite 80 flares reaching the surface mixed layer. This points towards oil coating bubbles collapsing in the upper sections of the water column and methane dissolving in the surface mixed layer.

### Implications for formerly glaciated hydrocarbon-bearing shelves

Our datasets from the Barents Sea document extensive oil and methane leakage from hydrocarbon reservoirs that have experienced uplift and glacial erosion, constituting an important source of methane to the water column, and, possibly, atmospheric inventory. Significantly, there are numerous analogous geological settings (sedimentary basins with petroleum potential that have experienced uplift and glacial erosion) across North Atlantic and Arctic continental

margins where we may expect abundant hydrocarbon leakage: the Timan-Pechora Basin in the Pechora Sea[30], the Sverdrup Basin on the northern margin of North America[63], the Eastern Basin in the Russian part of the Barents Sea[64], the western (Norwegian) part of the Barents Sea[40], sedimentary basins surrounding British Isles[65,66] and sedimentary basins of western and eastern Greenland[67–69] (Fig. 1). Indeed, landscape evolution modeling[36,70] suggests that several of these are more severely eroded than the Barents Sea shelf, for example, the eastern and western Greenland shelf, Canadian Arctic Archipelago and Norwegian Sea shelf, increasing the likelihood of reservoir seals being weakened or removed. However, studies of ongoing free gas release across these frontier basins are currently lacking. Expanded mapping and quantifying of seabed point-source emissions across high latitude glaciated shelves should be prioritised and may motivate rethinking of the contribution of thermogenic methane to global marine carbon sources.

## Methods

To find relationships between petroleum systems and natural hydrocarbon leakage we combined five sets of multibeam water column data (four CAGE data sets of scattered lines and one aerial MAREANO data set), a suite of 2D seismic lines and existing seismic interpretation results, as well as a series of SAR satellite images and discrete analyses of sea water and bottom sediments (Fig. S3). CAGE data sets were acquired in four research cruises onboard RV Helmer Hanssen: CAGE 18-1 cruise in May 2018, CAGE 19-2 cruise in July 2019, CAGE 20-2 cruise in July 2020, and CAGE 21-4 cruise in August 2021.

### Water column backscatter data acquisition

We used multibeam echosounder data to identify gas emission sites from the seafloor to the water column. Multibeam echosounders emit sound waves and measure the elapsed time for a wave to reach a

reflective object and return to the receiver (two-way travel time). The distance to the object is calculated based on a sound velocity profile. Because gas bubbles are excellent reflective targets due to their high acoustic impedance contrast with sea water[71], surveying water column with multibeam systems is an effective and reliable method for identifying submarine gas discharge sites[72,73].

All CAGE data sets were acquired with Kongsberg EM302 echosounder. The EM302 echosounder was operated with a 120° opening angle providing a swath of 432 soundings which covered an off-track area ~3 times the water depth. The ping rate was automatically adjusted depending on the water column thickness. A thicker water column increases the travel time of the acoustic signal and dictates longer intervals between pings. The datasets were acquired at 6-8 knot vessel speed. For sound velocity profile acquisition, we used SBE 911plus CTD sensor.

Publicly available MAREANO data sets were acquired by a third-party using Kongsberg EM710 multibeam system (see https://www.mareano.no/ for more details on data acquisition).

## Water column backscatter data interpretation
Acquired fan-shaped water column images (both, CAGE and MAREANO data sets) have distinctive side-lobe artefacts due to strong signal return from the seafloor within peripheral sectors of the swath (Fig. 3e, f, Fig. S5). The side-lobe artefacts appear outside of a half-circle (minimum slant range) with a radius equal to the shortest distance between the sonar and the seafloor (Fig. S5). The side-lobe artifacts are inevitable features of the multibeam water column data[73–75], which limit the data sector suitable for flare detection to ~50% of the total swath coverage (Fig. 3e, f, Fig. S5). This natural data limitation is broadly known in marine geophysics community[73–76].

The transmit fan of the Kongsberg EM302 is split in 4 sectors (Fig. S5) with independent steering to compensate for the vessel movements. Because the sectors have slight ping offsets, small static artefacts appear within the slant range limits (Fig. S5). The artefacts occur at a narrow depth interval from beam 1 to beam 33 and from beam 255 to beam 288 and do not significantly obstruct manual flare detection. MAREANO data acquired with Kongsberg EM710 have very minor static artifacts that do not hinder flare detection.

For gas seep mapping we used only 'observable water volume' where data quality allowed confident acoustic flare detection (Fig. S5). Within the observable volume of multibeam data, acoustic flares appear as clouds of strong scatter points. The position of each point is described by coordinates and water depth. We examined the entire volume of observable water in fan-view mode (Fig. 3f) in FMMidwater software by QPS and manually allocated (picked) coordinates of the visible flare roots. In cases of data overlap, we picked flares during the first passage only.

Comparing acoustic flares with in-situ observations of seabed gas release at the western Svalbard margin, Sahling et al.[54] showed that strong flares correspond to six clustered streams of bubbles on average, while weak flares typically represent a single bubble stream. Thus, acoustic flares may represent a wide range of gas emission strength, and counting them in an indiscriminate way may hinder spatial variability of the gas emissions. With the aim to map the activity of the gas release at detail, we opted for categorizing each acoustic flare within our datasets to weak, medium, and strong based on the signal scattering strength (Fig. 3f). Similar subjective assessment of the apparent flare magnitude has been previously done[77]. Aiming to ensure equability of data interpretation, all data were interpreted by one person and with constant colour range indicative for the range of signals and using the same screens. Random inspection of maximum scattering strength within flares has been also done. We assumed that strong flares have raw amplitudes exceeding 65, and weak flares demonstrate raw amplitude below 85. These values have been used for verification of flare strength interpretations throughout the data analysis.

Scattering strength may be the best available verification criteria as the dimensions of flares on water column images may depend on the oceanographic conditions (e.g., currents and stratification of water column) and varying beam-footprint width within the swath[78].

The picked flare locations were plotted as point maps and density maps (Figs. 3, 4). Density maps do not take into account the differences between weak, medium, and strong flares and rather show the concentration of individual flares as color-coded polygons. To produce the density maps we utilized interpolation method described by quartic Kernel function (1). Interpolation was done within search radius of 500 m.

$$K(y) = \frac{15}{16}(1-y^2)^2, |y| \le 1 \qquad (1)$$

## Oil slick detection on SAR images
Sentinel-1 Synthetic Aperture Radar (SAR) satellite images were downloaded from the Copernicus Open Access Hub (https://scihub.copernicus.eu/dhus/#/home). A selection of seven weather-compliant Sentinel-1 images from April to October 2020 was used in this study. Interpretation of oil slick outlines (Fig. 5c) was done manually in ArcGIS based on low backscatter areas with an oil slick appearance. The oil slick emission points (Fig. 5a) were determined based on the locations of all interpretations of oil slicks outlines in the three identified areas with large persistent oil slicks.

## 2D seismic acquisition and processing
CAGE seismic data were acquired using one GI (Generator-Injector) air gun as the seismic energy source. We used a 100-m long streamer with 32 channels separated by 3.125 m, which was composed of four consecutively connected 25 m-long P-Cable streamer sections[79]. Data processing was done in Radex Pro software and included CDP binning (3.125 × 3.125 m bin size), Simple Bandpass filtering (10–25–300–400 Hz) and Spherical Divergence, bubble removal, NMO correction to water velocity, zero-offset demultiple, migration using the post stack Kirchhoff migration algorithm. Output Seg-y data were interpreted in Petrel Software.

## Subbottom profiler (Chirp) data acquisition
The X-STAR Full Spectrum Sonar, transmitting a pulse linearly swept over a spectrum of frequencies (1.5 kHz to 9 kHz) and operated at 0.3 Hz ping rate, was used to image the shallow 0–30 m of subseafloor in detail. Acquired data did not need manual processing and allowed for distinguishing laminated soft marine sediments, glacigenic deposits, and lithified rocks (Fig. S1).

## Measurements of dissolved methane concentration in sea water
Water samples were collected with a CTD rosette and transferred into 120-ml glass bottles immediately after sampling. We added 1 ml of 1 M solution of NaOH into each bottle before capping them with a rubber septum and sealing with aluminium crimp caps. Before gas chromatographic analysis with flame ionization detector (GC-FID), 5 mL of laboratory pure $N_2$ gas was added to the bottles. GC-FID analyses were carried out using ThermoScientific Trace 1310 gas chromatograph.

## Measurements of hydrocarbon gas composition in bottom sediments
Sediment samples were collected with a gravity corer. For subtracting a gas phase from a sediment matrix, we used a conventional headspace gas sampling technique. 5 mL of bulk sediments were transferred into 20-mL glass vials containing 5 mL of 1 M solution of NaOH and two glass beads. Vials were capped with a rubber septum and sealed with crimp tops. Analyses were carried out using a ThermoScientific Trace

1310 gas chromatograph equipped with ThermoScientific TG-BOND alumnia ($Na_2SO_4$) 30 m x 0.53 mm×10 $\mu$m column.

## Sea-air methane flux calculations

The sea-air methane flux was determined using a bulk flux equation:

$$F = k(Cw - Co) \tag{2}$$

Where $k$ is gas transfer velocity (cm h$^{-1}$), $Co$ is methane concentration in the surface water in equilibrium with the overlying air (mol m$^{-3}$), $Cw$ is methane concentration measured in surface mixed layer (mol m$^{-3}$). The equilibrium concentration was calculated from the atmospheric pressure measured and the partial pressure of methane in dry air using the Bunsen solubility coefficient of methane at the temperature and salinity conditions of the seawater. Gas transfer velocity is estimated from the molecular diffusivity of methane gas and the kinematic viscosity of the seawater combined in Schmidt number ($Sc$), and the wind speed at 10 m above the sea surface ($u_{10}$)[80]:

$$k = 0.251 * u_{10}^2 \left(\frac{Sc}{660}\right)^{-0.5} \tag{3}$$

The relationship between the wind speed and the gas exchange (3) has 20% uncertainty[80]. $u_{10}$ was calculated from the wind speed measured during the water sampling ($u_{meas}$) at height of 22.4 m ($z_{meas}$)[81]

$$u_{10} = u_{meas} \left(\frac{Z_{meas}}{10}\right)^{-0.11} \tag{4}$$

## Data availability

The raw multibean echosubnder data for seafloor mapping and gas flare detection in the water column acquired by CAGE is available at https://dataverse.no/dataset.xhtml?persistentId=doi:10.18710/I3L0BQ[82]. The gas flare location data and oil slick emission point location data generated in this study are provided in the Source Data file. Seismic data from NPD – Norwegian Petroleum Directorate are available upon direct request. Sentinel −1 SAR images are available at https://scihub.copernicus.eu/. MAREANO multibeam echosounder data are available at https://www.mareano.no/en/maps-and-data/marine-geospatial-data. Source data are provided with this paper. IBCAO V4 bathymetric data (Figs. 1, 3,4,6,7) is available at https://www.gebco.net/data_and_products/gridded_bathymetry_data/arctic_ocean/ Source data are provided with this paper.

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

## Acknowledgements

We thank the crew of RV *Helmer Hanssen* for providing invaluable support for empirical data acquisition; Dr. Andreia Plaza-Faverola for leading CAGE 18–1 research cruise and providing multibeam data; Dr. Malin Johansson for useful discussions of remote sensing data; Frank Werner Jakobsen for providing photographs of oil slicks. The research is supported by the Research Council of Norway through its Centres of Excellence funding scheme (Grant 223259), through the large-scale interdisciplinary project ReGAME (Grant 325610), and by The Norwegian Academy of Science and Letters through its VISTA grant awarded to P.S.

## Author contributions

P.S. processed and interpreted multibeam echosounder data, drafted the manuscript, and designed the figures. R.M. interpreted satellite SAR images, provided conventional seismic data interpretation; M.W., and H.P., K.A. acquired multibeam echosounder, high-resolution seismic, oceanographic data; K.A., R.M., H.P., M.W., P.S. designed empirical data acquisition campaigns and outlined the scopes of the study. K.A., R.M., H.P., and M.W. contributed to manuscript writing and editing.

## Funding

## Competing interests

The authors declare no competing interests.
