## [Peer Review File · Nature Communications]

Widespread natural methane and oil leakage from sub-marine Arctic reservoirsReviewer #1 (Remarks to the Author):

The study by Serov et al. reports findings made of thermogenic methane (natural) seepage at the seafloor of the Arctic ocean. By the magnitude and distribution of these natural seeps, they represent an important finding to the scientific community, especially with respect to global methane budgets. The methods used appear robust, with several field campaigns, and are of interest to the scientific community.

I understand that this may partly be out of the scope of the current study, but as a reader I would be interested to learn more if anything can be said about both the fraction of the seeped methane reaching the atmosphere (and absolute amount). As a reader, I would be also interested to be provided with some information (maybe in the methods) if the authors considered the potential effect of tides (it has been found in the past that seepage might relate to tidal fluctuations at some locations whereas not at others; this might not be assessable for the current data set, but it is a question if such transient effects might potentially have affected the observed seepage distribution (was it important whether measurements at a given location were made at low versus high tide?)). Finally, because surface areas and number of seeps are provided, more information about what these quantities are would be warranted (e.g., investigated surface area (investigated with instruments) versus overall surface area of the region); the question also arises when the current authors compare a number of seeps they detected within a given surface area with number of seeps detected by previous authors within given surface areas: are the numbers really comparable(?). However, I do understand that the current study is already very detailed and focused on mapping the overall distribution of the seepage--the level of detail provided seems already appropriate, and these aspects might be discussed only very shortly (if at all), if appropriate/doable.

Additionally, I have a few more minor, line-by-line comments:

- line 23: would "widespread" be more accurate than "pervasive"?
- line 36: maybe remove "fugitive"; it is my understanding that some leakages may not be transient, as the use of this word might imply.
- line 37: consider adding "natural" before seepage?
- line 53: "contains" to "contain".
- line 98: add unit for " 940 to 1180".
- line 117 and throughout the manuscript: it would help if the authors could clarify what they mean with the surface areas provided. In a general area of X km², only Y km² were covered by the multibeam survey. This is clear from Figure 3, which also indicates that overlap happened (i.e., the authors measured several times at the same location). It would help to clarify a little these aspects, in the main text or in the methods.
- line 146: figure 3: weak, medium, and strong flares are difficult to differentiate when many flares overlap (points of same color but different sizes). The "high flares" are very visible, and this is maybe on purpose that the authors desired to focus the reader's attention on those.
- line 175 " intermediate water interval"; it may help to explain how the different intervals were defined.
- lines 180-181. "heavier methane homologs" is unclear. Do the authors refer to C₂-C₅ hydrocarbons (ethane, propane, etc.), or something similar? I am not used to the terms "methane homologs".
- lines 304-307: this effect is unlikely to be significant. As highlighted by the authors's figure 5d, the methane concentrations they observed remained much lower than the methane saturation aqueous solubility. I.e., the ambient methane concentrations are unlikely to substantially decrease the rate of aqueous dissolution of a gas bubble composed of initially close to 100% methane gas. This is actually what is stated by the reference 66 cited here by the authors (Gros et al., 2021), i.e. the authors cite this reference potentially incorrectly here. (See similar (but more detailed) reasoning presented by Gros et al. (2021), at lines 258-266 of their Supporting Information, where

they calculated quite negligible effect on bubble aqueous dissolution for 30 μM methane in water: the authors here seem to have observed substantially lower methane concentrations. (I mean, the authors are right that the model presented by Gros et al. would imply such an effect, if concentrations were large enough; however, Gros et al. also argued that it is unlikely to happen anywhere in the global oceans that concentrations large enough to substantially slow down aqueous dissolution of methane bubbles would be observed. And the current authors do not seem to show any concentration in the water column that would show that the analysis by Gros et al. was inaccurate.)

[Whereas reduced rate of mass transfer is unlikely to have a substantial effect at concentrations reported to date in seawater worldwide (as far as I am aware of), it is possible that large discharges of gas bubbles result in multiphase plumes (constituted of gas bubbles and entrained seawater), such that the overall ascent speed towards the sea surface is increased. This might need to be evaluated and might be a more significant effect than the effect described by the authors on aqueous dissolution rate.]

- line 311: this may be difficult in absence of measurement of bubble size, but discussion of the extent of methane emitted at the sea surface might have been of interest (see e.g. McGinnis et al., 2006, "Fate of rising methane bubbles in stratified waters: How much methane reaches the atmosphere?"). Though I understand that this might not be available with the data collected during this study. And the authors already give some information e.g. by their re-analysis of the measurements by Platt et al. (around line 315).

- line 323: "our methane concentration data collected within oil slicks"; unclear: are these concentration in water immediately below these slicks? Or in air? Or in the water at deeper water depth (which water depth)?

Reviewer #2 (Remarks to the Author):

The manuscript "Widespread natural methane and oil leakage from sub-marine Arctic reservoirs" presents a comprehensive analysis of seismic data, water column echosounder images of gas seepage, methane concentration in seawater from Norwegian Barents Sea, Arctic Ocean indicating an underestimated sources of methane and oil from hydrocarbon reservoirs to the ocean and possibly to the atmosphere. The response of hydrocarbon reservoirs to environmental changes (especially deglaciation) is of great relevance to predict feedback mechanisms during present-day global warming and thus highly relevant. The style and language of the manuscript is of high quality and the text is easy to follow.

I have one major comment, which I would like to be addressed by the authors. The main premise of the study is that all methane to the seawater originates from leaking petroleum-bearing sediments. One could assume that physical transport of methane to the surface may be the primary pathway; one could not rule out the local methanogenesis of the sea water. Multiple recent studies have repeatedly shown, methane production can and does occur under both anoxic and oxic conditions in the seas. Areas with widespread methane and oil leakage make such local methanogenesis even more likely. I wonder if the geochemical data of the methane in sea water is available, which is useful for the discrimination.

Point-by-point response

Reviewer #1: I understand that this may partly be out of the scope of the current study, but as a reader I would be interested to learn more if anything can be said about both the fraction of the seeped methane reaching the atmosphere (and absolute amount)

Authors: In the new version of the manuscript, we calculated bulk methane flux through the sea-air interface using the Wanninkhof method (lines 175-176, 317-319, 455-469). The flux varies from 1.2 to 3 $\mu\text{mol m}^{-2} \text{d}^{-1}$ across the 6 sampling locations. For comparison, at the shallower seep sites offshore Svalbard (90 - 240m water depth) the fluxes of 2 to 20 $\mu\text{mol m}^{-2} \text{d}^{-1}$ have been reported^{1,2}.

Calculating the total amount of methane reaching the atmosphere at our study area relying on few discrete measurements alone is problematic because neither the size of the methane plume in the water is known, nor the transient concentration gradients within it. All our water samples collected within 14 x 9 km rectangular area happen to be taken from one such plume as they demonstrate 19% to 35% CH₄ supersaturation. For comparison, a methane flux survey of Silyakova et al., 2020² contained 64 sampling stations within 22 x 10 km study site, which were repeated in 3 consecutive years. Given that the most active seep region in Sentralbanken spans over at least 620 km², more dense and repeated measurements of methane concentrations in the water are necessary. Our group has collected and will be collecting more water column and air samples, which will be used in a different paper.

Reviewer #1: “As a reader, I would be also interested to be provided with some information (maybe in the methods) if the authors considered the potential effect of tides (it has been found in the past that seepage might relate to tidal fluctuations at some locations whereas not at others; this might not be assessable for the current data set, but it is a question if such transient effects might potentially have affected the observed seepage distribution (was it important whether measurements at a given location were made at low versus high tide?))”

Authors: Indeed, tides cause reciprocal hydrostatic pressure perturbations in the shallow subsurface. Such fluctuations affect pressure-dependent solubility of gases in pore waters and may cause solution and exsolution of methane vapour phase. As Reviewer 1 points out, the strength of tidal effects on methane seepage changes significantly across seepage regions worldwide. This is likely due to the variable abundance of methane in pore water and variable solubility limits dictated by water depth and seafloor temperature at each region. If the amount of methane in seafloor is at or close to the solubility limit, small pressure decrease due to low tide may cause free gas release from otherwise ebullition-free seafloor³. However, if methane vapour phase is available within the seafloor in abundance and the ebullition is ongoing (e.g. at high tide), additional degassing at low tide is likely to be less noticeable on echosounder data.

We reanalysed seven circumstantially overlapping echosounder data sections acquired at different tidal settings to investigate whether tidal effects have a visible effect on the number of

observed gas flares. In the revised version of the manuscript we added a new Supplementary figure (Fig. S.1) showing tidal amplitude⁴ and number of gas flares within the repeatedly surveyed areas. Our analyses show that in three areas the number of flares increases at lower tide, in the three areas the number of flares decreases at lower tide, and in one area the number of flares remains the same regardless of tide settings. Thus, we do not observe a correlation between tide fluctuations and seep activity. We suggest that a potent thermogenic hydrocarbon sources at our study sites provide sustained and extensive gas flux through the seafloor significantly exceeding methane solubility limits and, thus, overriding detectable tidal solution-exsolution transient effects.

Reviewer #1: “the question also arises when the current authors compare a number of seeps they detected within a given surface area with number of seeps detected by previous authors within given surface areas: are the numbers really comparable(?). However, I do understand that the current study is already very detailed and focused on mapping the overall distribution of the seepage--the level of detail provided seems already appropriate, and these aspects might be discussed only very shortly (if at all), if appropriate/doable.”

Authors: We understand the concern of Reviewer 1 on comparability of gas seep mapping carried out using different echosounders. For flare mapping in Sentralbanken we used the same Kongsberg EM302 echosounder with the same opening angle as Skarke et al., 2014⁵ used to map flares on the US Atlantic margin. Therefore, we suggest that Skarke et al., 2014 and our study can be compared. On the western Svalbard margin, Mau et al., 2017⁶ used a different Kongsberg EM710 echosounder. Although EM302^{7,8} and EM710^{9,10} are very widely used to map acoustic gas flares, potential discrepancies of the backscatter water column strength records between them have not been documented in literature yet. Despite, we do not expect a significant difference in sensitivity to free gas in the water column between EM710 and EM302, we agree that the potential reader should be informed about differences in acquisition between Mau et al., 2017⁶ and our work (lines 265-266, 276-277).

Reviewer #1:- line 23: would "widespread" be more accurate than "pervasive"?

Authors: done

Reviewer #1:- line 36: maybe remove "fugitive"; it is my understanding that some leakages may not be transient, as the use of this word might imply.

Authors: done

Reviewer #1:- line 37: consider adding "natural" before seepage?

Authors: done

Reviewer #1:- line 53: "contains" to "contain".

Authors: done

Reviewer #1:- line 98: add unit for " 940 to 1180".

Authors: done

Reviewer #1:- line 117 and throughout the manuscript: it would help if the authors could clarify what they mean with the surface areas provided. In a general area of X km², only Y km² were covered by the multibeam survey. This is clear from Figure 3, which also indicates that overlap happened (i.e., the authors measured several times at the same location). It would help to clarify a little these aspects, in the main text or in the methods.

Authors: Thank you for pointing this out. Done.

Reviewer #1:- line 146: figure 3: weak, medium, and strong flares are difficult to differentiate when many flares overlap (points of same color but different sizes). The "high flares" are very visible, and this is maybe on purpose that the authors desired to focus the reader's attention on those.

Authors: due to the high density of flares, it is difficult to avoid overlapping symbols indicating flares. That is why we provided flare density maps (Figure 4). With Figure 3 we aim to show that the flares are abundant, and their types are mixed within the clusters (there are no flare clusters composed of only strong flares or only weak layers). "High flares" were purposely placed as the overlying map layer to make sure they are well visible as we specifically discuss them in the text.

Reviewer #1:- line 175 " intermediate water interval"; it may help to explain how the different intervals were defined.

Authors: here we considered a layer from 50 m above the seafloor to 50 m water depth. We added an explanation in line 181

Reviewer #1:- lines 180-181. "heavier methane homologs" is unclear. Do the authors refer to C2-C5 hydrocarbons (ethane, propane, etc.), or something similar? I am not used to the terms "methane homologs".

Authors: corrected

Reviewer #1:- lines 304-307: this effect is unlikely to be significant. As highlighted by the authors's figure 5d, the methane concentrations they observed remained much lower than the methane saturation aqueous solubility. I.e., the ambient methane concentrations are unlikely to substantially decrease the rate of aqueous dissolution of a gas bubble composed of initially close to 100% methane gas. This is actually what is stated by the reference 66 cited here by the authors (Gros et al., 2021), i.e.

the authors cite this reference potentially incorrectly here. (See similar (but more detailed) reasoning presented by Gros et al. (2021), at lines 258-266 of their Supporting Information, where they calculated quite negligible effect on bubble aqueous dissolution for 30 μM methane in water: the authors here seem to have observed substantially lower methane concentrations. (I mean, the authors are right that the model presented by Gros et al. would imply such an effect, if concentrations were large enough; however, Gros et al. also argued that it is unlikely to happen anywhere in the global oceans that concentrations large enough to substantially slow down aqueous dissolution of methane bubbles would be observed. And the current authors do not seem to show any concentration in the water column that would show that the analysis by Gros et al. was inaccurate.)

[Whereas reduced rate of mass transfer is unlikely to have a substantial effect at concentrations reported to date in seawater worldwide (as far as I am aware of), it is possible that large discharges of gas bubbles result in multiphase plumes (constituted of gas bubbles and entrained seawater), such that the overall ascent speed towards the sea surface is increased. This might need to be evaluated and might be a more significant effect than the effect described by the authors on aqueous dissolution rate.]

Authors: thank you for pointing this out. We revisited Gros et al., paper and agree on the Reviewer 1 comment. We removed the discussion of the reduced mass transfer between the bubbles and the water due to somewhat elevated dissolved methane content.

Reviewer #1:- line 311: this may be difficult in absence of measurement of bubble size, but discussion of the extent of methane emitted at the sea surface might have been of interest (see e.g. McGinnis et al., 2006, "Fate of rising methane bubbles in stratified waters: How much methane reaches the atmosphere?"). Though I understand that this might not be available with the data collected during this study. And the authors already give some information e.g. by their re-analysis of the measurements by Platt et al. (around line 315).

Authors: Bubble size measurements are unfortunately not available. Nevertheless, we added a few sentences on the fate of methane bubbles in the water column (e.g. mass transfer between the gas bubbles and the surrounding water) (lines 337-340).

Reviewer #1:- line 323: "our methane concentration data collected within oil slicks"; unclear: are these concentration in water immediately below these slicks? Or in air? Or in the water at deeper water depth (which water depth)?

Authors: It was collected 5 m below the slicks. We clarified it in line 334.

Reviewer #2: “I have one major comment, which I would like to be addressed by the authors. The main premise of the study is that all methane to the seawater originates from leaking petroleum-bearing sediments. One could assume that physical transport of methane to the surface may be the primary pathway; one could not rule out the local methanogenesis of the sea water. Multiple recent studies have repeatedly shown, methane production can and does occur under both anoxic and oxic conditions in the seas. Areas with widespread methane and oil leakage make such local methanogenesis even more likely. I wonder if the geochemical data of the methane in sea water is available, which is useful for the discrimination.”

Authors: Indeed, some methanogens have been shown tolerant to oxygen and *in vitro* studies proposed biochemical pathways of methanogenesis in sea water^{11,12}. However, understanding whether these pathways may generate sizeable sea water methane plumes is currently lacking. The majority of studies concerning methanogenesis in oxic conditions has been carried out in lacustrine settings.

Discriminating methane generated in the sea water at our study site requires analysis of isotopic composition of carbon and hydrogen of dissolved methane coupled with nutrient analysis and biogeochemical experiments of isolated samples. Unfortunately, we do not have equipment, expertise and sample material to carry out such biogeochemical investigations. Given lack of geochemical data to discriminate potential methane production in oxic conditions at our study sites, we agree with Reviewer 2 that we cannot rule out a possibility that a fraction of gas dissolved in the water at Sentralbanken area may originate from local methanogenesis in sea water (lines 185-187)

Methanogenesis in sea water can be ruled out as a contributor to free gas phase with certainty because low rates of local methane production (a range between 0.04 and 0.23 mmol m⁻³ day⁻¹ has been reported for lake environments^{11,13}) are incompatible with generating methane at quantities required to form gas bubbles.

Reference

- 1 Graves, C. A. *et al.* Fluxes and fate of dissolved methane released at the seafloor at the landward limit of the gas hydrate stability zone offshore western Svalbard. *Journal of Geophysical Research - Oceans* **120**, 6185-6201 (2015). <https://doi.org/10.1002/2015jc011084>
- 2 Silyakova, A. *et al.* Physical controls of dynamics of methane venting from a shallow seep area west of Svalbard. *Continental Shelf Research* **194**, 104030 (2020). <https://doi.org/10.1016/j.csr.2019.104030>
- 3 Sultan, N., Plaza-Faverola, A., Vadakkepuliambatta, S., Buenz, S. & Knies, J. Impact of tides and sea-level on deep-sea Arctic methane emissions. *Nature Communications* **11**, 5087 (2020). <https://doi.org/10.1038/s41467-020-18899-3>
- 4 Egbert, G. D. & Erofeeva, S. Y. Efficient inverse modeling of barotropic ocean tides. *Journal of Atmospheric and Oceanic technology* **19**, 183-204 (2002).

- 5 Skarke, A., Ruppel, C., Kodis, M., Brothers, D. & Lobecker, E. Widespread methane leakage from the sea floor on the northern US Atlantic margin. *Nature Geoscience* **7**, 657-661 (2014). <https://doi.org:10.1038/ngeo2232>
- 6 Mau, S. *et al.* Widespread methane seepage along the continental margin off Svalbard - from Bjørnøya to Kongsfjorden. *Scientific Reports* **7**, 42997 (2017). <https://doi.org:10.1038/srep42997>
- 7 Urban, P., Köser, K. & Greinert, J. Processing of multibeam water column image data for automated bubble/seep detection and repeated mapping. *Limnology and Oceanography: Methods* **15**, 1-21 (2017). <https://doi.org:https://doi.org/10.1002/lom3.10138>
- 8 Dupré, S. *et al.* Tectonic and sedimentary controls on widespread gas emissions in the Sea of Marmara: Results from systematic, shipborne multibeam echo sounder water column imaging. *Journal of Geophysical Research: Solid Earth* **120**, 2891-2912 (2015). <https://doi.org:https://doi.org/10.1002/2014JB011617>
- 9 Römer, M. *et al.* Assessing marine gas emission activity and contribution to the atmospheric methane inventory: A multidisciplinary approach from the Dutch Dogger Bank seep area (North Sea). *Geochemistry, Geophysics, Geosystems* **18**, 2617-2633 (2017). <https://doi.org:https://doi.org/10.1002/2017GC006995>
- 10 Colbo, K., Ross, T., Brown, C. & Weber, T. A review of oceanographic applications of water column data from multibeam echosounders. *Estuarine, Coastal and Shelf Science* **145**, 41-56 (2014). <https://doi.org:https://doi.org/10.1016/j.ecss.2014.04.002>
- 11 Tang, K. W., McGinnis, D. F., Frindte, K., Brüchert, V. & Grossart, H.-P. Paradox reconsidered: Methane oversaturation in well-oxygenated lake waters. *Limnology and Oceanography* **59**, 275-284 (2014). <https://doi.org:https://doi.org/10.4319/lo.2014.59.1.0275>
- 12 Karl, D. M. *et al.* Aerobic production of methane in the sea. *Nature Geoscience* **1**, 473-478 (2008). <https://doi.org:10.1038/ngeo234>
- 13 Bogard, M. J. *et al.* Oxic water column methanogenesis as a major component of aquatic CH₄ fluxes. *Nature Communications* **5**, 5350 (2014). <https://doi.org:10.1038/ncomms6350>

Reviewer #1 (Remarks to the Author):

The authors satisfactorily attended the comments by the two reviewers.

I do have one remaining comment. In the response by the authors, I read:

Reviewer #1: "the question also arises when the current authors compare a number of seeps they detected within a given surface area with number of seeps detected by previous authors within given surface areas: are the numbers really comparable(?). However, I do understand that the current

study is already very detailed and focused on mapping the overall distribution of the seepage--the level

of detail provided seems already appropriate, and these aspects might be discussed only very shortly (if at all), if appropriate/doable."

Authors: We understand the concern of Reviewer 1 on comparability of gas seep mapping carried out using different echosounders. For flare mapping in Sentralbanken we used the same Kongsberg EM302 echosounder with the same opening angle as Skarke et al., 2014 used to map flares on the US Atlantic margin. Therefore, we suggest that Skarke et al., 2014 and our study can be compared. On the western Svalbard margin, Mau et al., 2017 used a different Kongsberg EM710 echosounder. Although EM302,8 and EM710,10 are very widely used to map acoustic gas flares, potential discrepancies of the backscatter water column strength records between them have not been

documented in literature yet. Despite, we do not expect a significant difference in sensitivity to free gas in the water column between EM710 and EM302, we agree that the potential reader should be informed about differences in acquisition between Mau et al., 2017

and our work (lines 265-266, 276-277).

 My comment is the following: comparability of the instruments used is one important aspect, indeed. However, one should further note that any study usually proceeds along lines where gas seeps are detected (e.g., along the ship track). This means usually that not 100% but rather a much lower percentage of the total area of the studied region is actually investigated for seeps. Seep occurring outside of the ship tracks may remain undetected. Hence it makes comparisons between studies difficult, unless there is a clear description of what the reported areas are (area of seafloor

covered by the instrument or overall area of the region?). For example, line 274 mentions 570 seeps within 94,000 km². Lines 117-118 state that the study by the authors of the manuscript had a seafloor coverage of 660 km². The reader can probably assume that the given surface area are comparable, but from the current text a doubt remains for the reader whether the surface areas provided are comparable or not.

 This remains however a detail that might not need any text edit or small text edits that could probably be included at the stage of proof corrections.

Reviewer 1: My comment is the following: comparability of the instruments used is one important aspect, indeed. However, one should further note that any study usually proceeds along lines where gas seeps are detected (e.g., along the ship track). This means usually that not 100% but rather a much lower percentage of the total area of the studied region is actually investigated for seeps. Seep occurring outside of the ship tracks may remain undetected. Hence it makes comparisons between studies difficult, unless there is a clear description of what the reported areas are (area of seafloor covered by the instrument or overall area of the region?). For example, line 274 mentions 570 seeps within 94,000 km². Lines 117-118 state that the study by the authors of the manuscript had a seafloor coverage of 660 km². The reader can probably assume that the given surface area are comparable, but from the current text a doubt remains for the reader whether the surface areas provided are comparable or not.

Authors: we agree with the comment made by Reviewer 1 that, unless the exact insonified area is reported, the uncertainty in the total number of detected seeps remains. This is indeed the case in the gas seep mapping on the western Svalbard margin by Mau et al., 2017¹. We made the appropriate changes in the manuscript. However, Skarke et al., 2014² clearly state “Here we use multibeam water-column backscatter data that cover 94,000 km² of sea floor...”. The data used in this paper is available at the Water Column Sonar Database (<https://www.ncei.noaa.gov/maps/water-column-sonar/>) and we can confirm the data coverage (not the total study area size) is 94,000km². We appreciate the concern of the Reviewer 1 as clear data coverage information is important when comparing gas release activity across different regions.

- 1 Mau, S. *et al.* Widespread methane seepage along the continental margin off Svalbard - from Bjørnøya to Kongsfjorden. *Scientific Reports* **7**, 42997 (2017).
<https://doi.org:10.1038/srep42997>
- 2 Skarke, A., Ruppel, C., Kodis, M., Brothers, D. & Lobecker, E. Widespread methane leakage from the sea floor on the northern US Atlantic margin. *Nature Geoscience* **7**, 657-661 (2014).
<https://doi.org:10.1038/ngeo2232>